# Effect of a Passive Exosuit on Sit-to-Stand Performance in Geriatric Patients Measured by Body-Worn Sensors—A Pilot Study

**DOI:** 10.3390/s23021032

**Published:** 2023-01-16

**Authors:** Ulrich Lindemann, Jana Krespach, Urban Daub, Marc Schneider, Kim S. Sczuka, Jochen Klenk

**Affiliations:** 1Department of Clinical Gerontology, Robert-Bosch-Hospital, 70376 Stuttgart, Germany; 2Biomechatronic Systems, Fraunhofer Institute for Manufacturing Engineering and Automation IPA, 70569 Stuttgart, Germany; 3Institute of Epidemiology and Medical Biometry, Ulm University, 89081 Ulm, Germany; 4Study Center Stuttgart, IB University of Health and Social Sciences, 70178 Stuttgart, Germany

**Keywords:** exosuit, sit-to-stand, peak angular velocity, geriatric patient

## Abstract

Standing up from a seated position is a prerequisite for any kind of physical mobility but many older persons have problems with the sit-to-stand (STS) transfer. There are several exosuits available for industrial work, which might be adapted to the needs of older persons to support STS transfers. However, objective measures to quantify and evaluate such systems are needed. The aim of this study was to quantify the possible support of an exosuit during the STS transfer of geriatric patients. Twenty-one geriatric patients with a median age of 82 years (1.–3.Q. 79–84 years) stood up at a normal pace (1) from a chair without using armrests, (2) with using armrests and (3) from a bed with pushing off, each condition with and without wearing an exosuit. Peak angular velocity of the thighs was measured by body-worn sensors. It was higher when standing up with exosuit support from a bed (92.6 (1.–3.Q. 84.3–116.2)°/s versus 79.7 (1.–3.Q. 74.6–98.2)°/s; *p* = 0.014) and from a chair with armrests (92.9 (1.–3.Q. 78.3–113.0)°/s versus 77.8 (1.–3.Q. 59.3–100.7)°/s; *p* = 0.089) compared to no support. There was no effect of the exosuit when standing up from a chair without using armrests. In general, it was possible to quantify the support of the exosuit using sensor-measured peak angular velocity. These results suggest that depending on the STS condition, an exosuit can support older persons during the STS transfer.

## 1. Introduction

The capacity to stand up from a seated position is a prerequisite for independence and any kind of physical mobility. Unfortunately, more than 40% of older persons have problems with the sit-to-stand (STS) transfer [1], which requires 2.1 W/Kg and 2.6 W/Kg STS power in women and men, respectively [2]. As a consequence, poor performance during STS transfers is associated with falls in frail older persons living in nursing homes [3] and in community-dwelling older adults [4]. A major factor related to loss of mobility in older persons is sarcopenia, which is defined as a progressive and generalized loss of skeletal muscle mass and strength, mostly as a result of ageing [5]. Therefore, strategies should be developed to improve the STS transfer by training on a muscular and functional basis, by optimizing the STS context and/or through external support.

Strength and balance training is the recommended standard to improve the STS transfer in older persons [6,7]. It is also one aspect of fall prevention in frail older persons [8,9]. Using armrests and individually adapting the seat height and foot position are strategies to optimize the STS context [10], and should be part of physiotherapy and occupational therapy. External support could be provided by exoskeletons, which support older persons transferring from sitting to standing. Currently, lower-limb exoskeletons are available only for gait rehabilitation, but they are still in their early stages of development [11]. In contrast, exoskeletons for STS support are not available at all. Any technical support should take into account the mechanical and metabolic burden (i.e., weight) of the device [12] resulting in a trade-off between compliance and support. Exosuits are a specific type of exoskeleton without a hard rigid structure for power transmission. Although in exosuits the external support of the movement is limited, the weight is minimal and possible joint misalignment due to rigid material [13] is avoided. A passive assistive device based on elastic straps was developed to support knee extension [14], but not to support the total chain of body extension (knee, hip, trunk), which is necessary during the STS transfer. There are several exosuits available for industrial work, i.e., lifting heavy weights [15]. Since the STS transfer involves the same sequence of movements, such as forward trunk lean followed by leg and trunk extension, these products might be adapted to the needs of older persons to support and improve STS transfers. To quantify and evaluate the support of an exosuit during STS transfer, sensor technology could be used.

As an outcome of STS transfer performance, the digital assessment of a single STS transfer or several transfers at a normal pace during daily life is more ecologically valid than repeating the STS transfer several times as fast as possible, such as the 5-Chair-Rise test [16]. The digital assessment of STS performance can be conducted by locally tied devices in a laboratory, such as opto-electronic systems [17] or force plates [18]. In contrast, wearable sensors, such as accelerometers and gyroscopes, have the potential to describe the STS transfer quantitatively during daily life [19]. Here, the peak angular velocity (PAV) of the thigh is a relevant and reliable parameter to describe STS performance in a laboratory [20] and during daily-life activity monitoring [21].

The aim of this study was to quantify the possible support of an exosuit during the STS transfer of geriatric patients. We hypothesized that thigh PAV is faster during the STS transfer when wearing an exosuit compared to not wearing.

## 2. Materials and Methods

### 2.1. Subjects and Design

For this experimental study with cross-sectional design, geriatric patients of a south-west German geriatric rehabilitation clinic were recruited. Inclusion criteria were an age of 60 years or older and problems with the STS transfer according to the senior physician in charge. Exclusion criteria were major balance impairment (self-report), cognitive impairment (according to the senior physician in charge), acute pain during the STS transfer (self-report) and terminal illness. The study protocol was approved by the ethical committee of the University of Tübingen (653/2021BO1) and all participants gave written informed consent.

### 2.2. Protocol and Outcome Parameters

Participants were instructed to stand up at normal pace (1) from a chair of 46 cm seat height without using armrests, (2) with using armrests and (3) from a bed of same height with pushing off from the mattress. A second chair was placed in front of the participant for safety, but the participants were instructed to hold on only in case of acute balance problems during or after the STS transfer. Position of the feet was standardized in a parallel position [10]. STS transfers were performed while wearing a passive exosuit (Rakunie RK2, Morita Group, Tokyo, Japan; Figure 1) or without it. The exosuit was worn similar to a jacket and was closed in front of the chest with a strap. Attached elastic straps were leading down on the person’s back and were fixed with Velcro straps below the knees. The supportive force of the exosuit is created by the elastic bands at the body’s back and buttocks that are stretched when sitting down. The stored energy in the elastic bands is released during the STS transfer. The intensity of the elastic bands can be adjusted via loops by means of Velcro straps but cannot be scaled reproducibly regarding maximum output force. Therefore, the intensity was adjusted individually so that the patient felt a clear tension while standing upright with the same tension at each trial. Two trials were performed at each condition (in total 12 STS transfers) in randomized order to account for a learning effect or fatigue. The first trial was a rehearsal trial, and the second trial was used for analysis.

### 2.3. Outcome Measure

Two inertial measurement units (IMUs) (xSens DOT, xSens Technologies B.V., Enschede, The Netherlands) were fixed by Velcro straps at the front of the participants’ thighs. Angular velocity of the thighs was recorded by gyroscopes at 120 Hz. After lowpass filtering with a 3rd order Butterworth filter with a cut-off frequency of 6 Hz, mean PAV of the right and left thigh was calculated and used as the outcome parameter to describe STS performance [20]. After all measurements, the usability of the exosuit was assessed with the System Usability Score [22], a 10-question interview scoring 0 (worst) to 100 (best).

### 2.4. Descriptive Measures

Age, gender, body weight and body height were used to describe the cohort. Furthermore, comorbidity was assessed via a standardized questionnaire [23] asking for 18 diseases and symptoms. Yes/no answers resulted in a (0–18) sum-score. Cognition was screened via the Short Orientation Memory Concentration test [24]. It assesses temporal orientation, counting backwards, recounting the months and short-term memory, resulting in a weighted score of possible 0–28 errors. Habitual gait speed over 8 m was measured by a stopwatch and was used as a descriptive measure of general functional performance [25]. Handgrip strength was measured with a dynamometer (Jamar Smart Hand Dynamometer, Patterson Medical Ltd., Chicago, IL, USA). Mean (left/right) strength was used for analysis.

### 2.5. Statistics

Due to the uneven distribution of some parameters, non-parametric statistical analyses were applied. Group median values with 1st and 3rd quartile (1.–3.Q), minimum and maximum, were used to describe outcome parameters. Wilcoxon signed rank tests were used to analyze differences between conditions. Spearman’s coefficient of correlation was calculated to describe the association between parameters.

Filtered sensor signals were visualized with the same length by interpolation. The Dynamic Time Warping Barycenter Average method [26] was applied to average the time series. Dynamic Time Warping compensates differences between sensor signals by allowing an elastic shift of the time axis [27]. This method was successfully applied before for recognition of upper limb movement patterns [28] and locomotion, such as walking and stair climbing [29].

## 3. Results

Twenty-one geriatric patients with a median age of 82 years (1.–3.Q: 79–84 years) were investigated, including 13 women (62%). Participants’ disease sub-groups were 16 patients (76%) with a generalized decompensation, 3 patients (14%) with orthopedic problems and 2 patients (10%) with neurologic problems. Most of them (n = 16; 76%) used a wheeled walker; all others did not use a walking aid. The characteristics of the cohort are described in detail in Table 1.

One patient was not able to stand up from the bed. In those who were able to stand up from the bed, the median PAV was statistically significant faster during the STS transfer from bed when using the exosuit compared to not wearing it (*p* = 0.014) with a median difference of 14.4°/s (1.–3.Q—6.3–28.0°/s). Single and best fit averaged gyroscopic sensor signals are shown in Figure 2. The median PAV was also faster during the STS transfer from a chair with armrest support when using the exosuit compared to not wearing it (*p* = 0.089). The median difference was 12.8°/s (1.–3.Q—2.3–25.1°/s). Three patients were not able to stand up from the chair without using armrests. In those who were able to stand up from the chair without using armrests, median PAV was not statistically significant between both groups (*p* = 0.966). The results are described in detail in Table 2 and are shown in Figure 3.

Comparing the averaged sensor signals from all conditions show that the main effect of the exosuit is during the first phase of the STS transfer (Figure 4).

The coefficients of correlation between differences in PAV with and without exosuit support with gait speed or hand grip strength did not show a clear pattern in all conditions and were all below |0.4|. The coefficients of correlation are shown in detail in Table 3.

After the measurements, the median System Usability Score was 66.3 (1.–3.Q. 57.5–71.3), ranging from 30 (minimum) to 95 (maximum).

## 4. Discussion

The results of this pilot-study support our hypothesis that using an exosuit improves STS performance by increasing PAV of the thighs during the STS transfer up to 19% at least when arms are used additionally. It was possible to quantify the supportive effect of the system using wearable sensor technology. The group of patients was well-selected because they had a low walking speed indicating poor physical performance and, according to the senior physician in charge, had problems getting up from the chair. PAV values in this study including only geriatric patients are reasonable but lower than in our previous proof-of-concept study including young healthy persons, older adults and geriatric patients with a mean PAV of 124.6°/s [20]. Since we measured thigh PAV in a supervised condition (laboratory), this can explain that PAV values were slightly faster than in the study that measured thigh PAV during daily life in healthy volunteers with a mean PAV of 70.7°/s [21].

For different test conditions, different results were found. When standing up from a bed, only little support by pushing off with hands is possible because arms are already nearly stretched when starting the movement. Therefore, the statistically significant result is reasonable assuming an exosuit effect. In contrast, when standing up from a chair using armrests, the arm support is more intensive and could have weakened the effect of the exosuit. Even worse, the exosuit seems to decelerate the STS transfer when standing up from a chair without using armrests. Here, the person has to create a fast forward momentum of the trunk by accelerated hip flexion [30]. This fast momentum is confirmed by faster PAV values in this condition without using the exosuit, but this movement is likely to be blocked by the exosuit’s elastic bands at the body’s back, which are stretched when the hip is flexed. Therefore, this result according to PAV change is reasonable. The limitation might not be clinically relevant since those persons who usually stand up from a chair without using armrests do not need external support.

There was no clear pattern describing the association between changes in PAV and classical assessment parameters of physical performance (gait speed and hand grip strength). Based on our data, we cannot tell if physically high or low performers benefit most from using this exosuit. Future studies should investigate more direct measures of lower limb functional performance for explanation.

Our results support the findings of Lee et al. [14], who found that a passive assistive device reinforcing knee extension was able to facilitate the STS transfer by reducing metabolic cost in healthy younger persons. Since, in general, the total chain of body extension (knee, hip, trunk) is necessary during the STS transfer, an exosuit supporting the total chain could have facilitated the STS transfer even more. When using the device for several days or longer, the supportive effect may even increase, since a familiarization effect is described for tolerating external forces [31].

Exosuits could be integrated into physiotherapy when STS exercises are limited by weakness or pain. Using an exosuit during daily life might be possible if other activities, e.g., toileting, are intuitively possible while wearing the device. With the current design, this is not feasible. Our results of the usability rating are lower than scores rating usability of robotic upper limb rehabilitation systems [32,33] and, thus, showing room for improvement. Here, further development of the exosuit is necessary.

It was possible to use PAV to quantify the effect of the exosuit under different conditions. It is a cheap and clinically applicable method. Future exosuits might directly provide embedded IMUs. The data can also be used during therapy to better adapt the system and as a biofeedback. Furthermore, long-term monitoring of activities while using the exosuit would be possible. Besides the thigh location, additional IMUs at the trunk might help to better understand the complex movements and support of the different body segments involved during the STS transfer.

In this study, we have focused only on the STS transfer but not on walking, because the elastic bands are not stretched during walking with this kind of exosuit. Future studies should investigate possible effects of soft exoskeletons/exosuits on walking performance.

Although the sample size was small, we could find relatively strong and statistically significant effects. However, the results must be reproduced in a larger study. A limitation of the study is that the intensity of the elastic bands could not be scaled reproducibly. The integration of strain gauges into the elastic bands and pressure sensors to quantify the pressure/uncomfortableness on the body could be a further development for scientific use.

## 5. Conclusions

In conclusion, our results suggest that it is possible to use wearable IMUs to quantify the effect of an exosuit on STS transfers in older people. Additional IMUs at the trunk might further improve the results. Depending on the STS condition, an exosuit can support older persons during the STS transfer, but further development is necessary regarding all day usability.

## Figures and Tables

**Figure 1 sensors-23-01032-f001:**
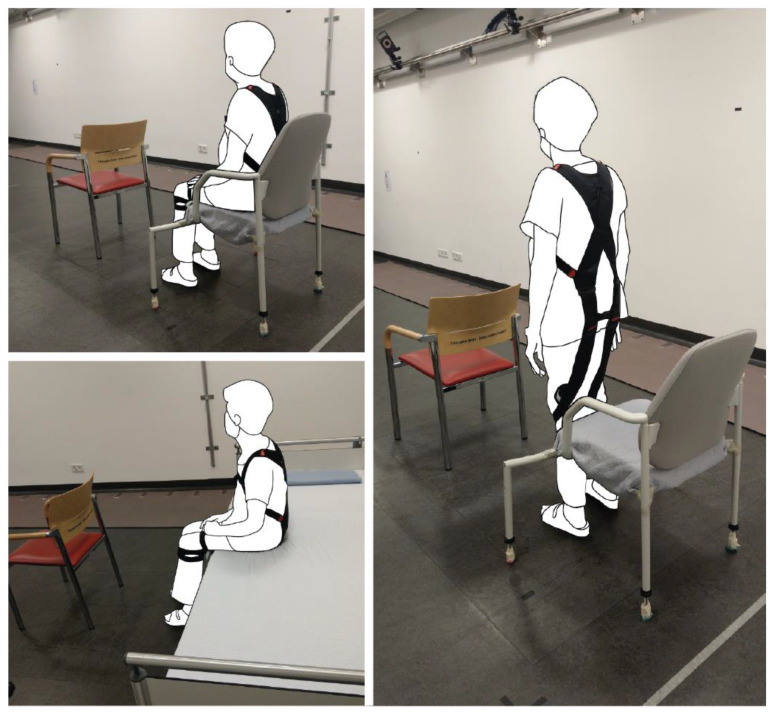
Participant equipped with a passive exosuit while sitting on a chair of 46 cm height, after standing up from a chair and while sitting on a bed of the same height.

**Figure 2 sensors-23-01032-f002:**
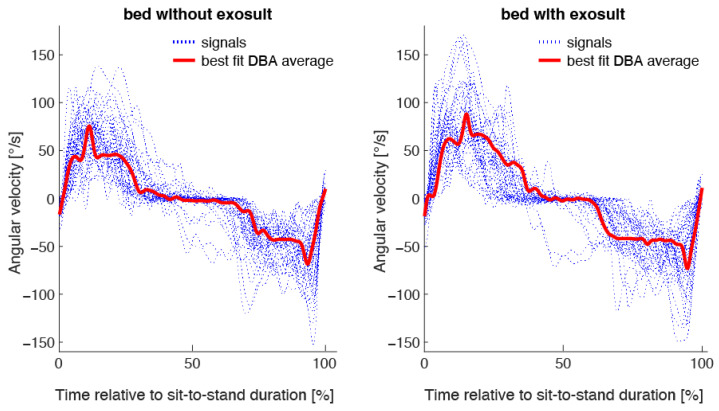
Single gyroscopic sensor signals adjusted to the same length by interpolation and best fit Dynamic Time Warping Barycenter Average (DBA) for sit-to-stand transfers from bed without and with using an exosuit. Time was standardized relative to the sit-to-stand duration.

**Figure 3 sensors-23-01032-f003:**
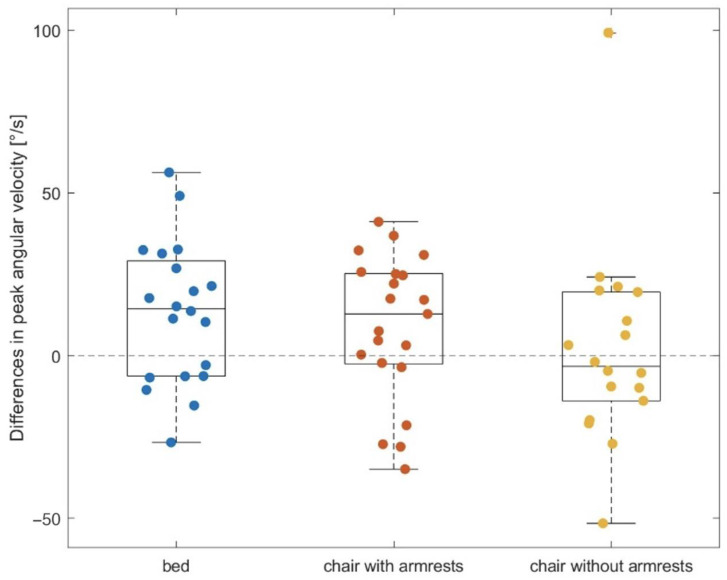
Boxplots and single measurements for differences in peak angular velocity between using an exosuit or not during sit-to-stand transfer from bed, chair with armrest support and chair without armrest support.

**Figure 4 sensors-23-01032-f004:**
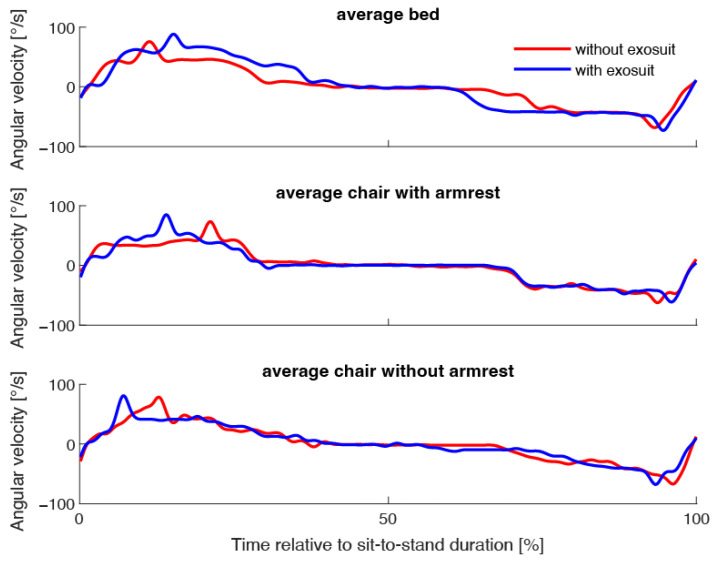
Best fit Dynamic Time Warping Barycenter averages for sit-to-stand transfers from bed, chair with armrests and chair without armrests, without and with using an exosuit. Time was standardized relative to the sit-to-stand duration.

**Table 1 sensors-23-01032-t001:** Characteristics of all participating geriatric patients (n = 21), including 13 (62%) women.

	Median (1.–3.Q)	Minimum–Maximum
Age (years)	82 (79–84)	64–92
Body weight (kg)	71.0 (60.0–87.0)	40.5–104.0
Body height (m)	1.66 (1.62–1.76)	1.45–1.90
Comorbidity index (0–18)	3 (3–4)	2–6
Cognition (0–25)	4 (2–6.5)	0–12
Habitual gait speed (m/s)	0.70 (0.51–0.79)	0.41–1.17
Maximum handgrip strength (kg)	20.0 (17.8–29.6)	14.4–51.0

1.–3.Q = 1st to 3rd Quartile; comorbidity was assessed via the Functional Comorbidity Index; cognition was screened via the Short Orientation Memory Concentration test; note: better score values are underlined.

**Table 2 sensors-23-01032-t002:** Median peak angular velocity of the right and left thighs during the sit-to-stand transfer of 21 geriatric patients.

Peak Angular Velocity (°/s) While Standing Up from …	Median (1.–3.Q)	Min–Max	*p*
bed with push off without exosuit *	79.7 (74.6–98.2)	56.0–140.2	0.014
bed with push off with exosuit *	92.6 (84.3–116.2)	50.6–164.6	
chair using armrests without exosuit	77.8 (59.3–100.7)	31.5–160.6	0.089
chair using armrests with exosuit	92.9 (78.3–113.0)	29.3–163.2	
chair without using armrests without exosuit **	86.8 (68.8–114.3)	35.4–147.4	0.966
chair without using armrests with exosuit **	83.7 (73.4–112.4)	33.5–204.8	

1.–3.Q = 1st quartile to 3rd quartile; *p* = probability; * n = 20; ** n = 18.

**Table 3 sensors-23-01032-t003:** Spearman’s coefficients of correlation between differences in peak angular velocity with and without exosuit support with gait speed or hand grip strength when standing up from bed or chair with or without armrests.

Differences in Peak Angular Velocity with and without Exosuit Support When …	Gait Speed	Hand Grip Strength
standing up from bed with pushing off	r = 0.33	r = 0.07
standing up from chair using armrests	r = 0.02	r = 0.31
standing up from chair without using armrests	r = −0.13	r = −0.38

## Data Availability

Data are available on request from the authors.

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
