# Peer review of "Effect of a Passive Exosuit on Sit-to-Stand Performance in Geriatric Patients Measured by Body-Worn Sensors—A Pilot Study"

_sensors, 2023, doi:10.3390/s23021032_

Round 1
Reviewer 1 Report
This paper presents an effective method (using accelerometers and gyroscopes to measure the peak angular velocity of the thigh) to quantify the possible support of an existing passive exosuit during the sit-to-stand transfer of geriatric patients. Additionally, three different conditions are selected to simulate actual application scenarios, and twenty-one geriatric patients are recruited to perform measurements. Finally, corresponding experimental results support the authors’ hypothesis that using a passive exosuit improves the sit-to-stand performance by increasing the peak angular velocity of the thigh. Although the research is very significant, the paper needs improvement before publication:
1. In the study, participants are instructed to stand up under three conditions: (1) from a chair of 46 cm seat height without using armrests, (2) with using armrests and (3) from a bed of the same height. It is higher when standing up with exosuit support under the second and third conditions. Why does not the peak angular velocity of the thigh increase under the first condition? I guess the first is able to verify that the passive exosuit can help geriatric patients restore their standing ability.
2. See page 3, line 89, the authors mentioned a passive exosuit. Is it a product? Can you include the relevant parameters in detail, such as stiffness, maximum output force, and so on?
3. When subjects wearing the passive exosuit are sitting, they may be feeling uncomfortable due to squeezing pressure from the elastic bands. Can you quantify the pressure? Furthermore, is there any evaluation standard indicating that the squeezing pressure is within a reasonable range?
4. Can the opto-electronic systems be used to measure the peak angular velocity of the thigh to reflect the reliability of the inertial sensing method?
5. I think the statement of Conclusion part is too short to achieve a comprehensive summary.
Author Response
Thank you for your time and effort to read our manuscript. Your comments were very helpful to improve the manuscript.
Point 1: In the study, participants are instructed to stand up under three conditions: (1) from a chair of 46 cm seat height without using armrests, (2) with using armrests and (3) from a bed of the same height. It is higher when standing up with exosuit support under the second and third conditions. Why does not the peak angular velocity of the thigh increase under the first condition? I guess the first is able to verify that the passive exosuit can help geriatric patients restore their standing ability.
Response: The seat-height of the three condition was the same. We have tried to explain why the exosuit was not helpful in condition (1):
“… the exosuit seems to decelerate the STS transfer when standing up from a chair without using armrests. Here, the person has to create a fast forward momentum of the trunk by accelerated hip flexion [30]. This fast momentum is confirmed by faster PAV values in this condition without using the exosuit, but this movement is likely to be blocked by the exosuit´s elastic bands at the body´s back, which are stretched when the hip is flexed.”
Point 2: See page 3, line 89, the authors mentioned a passive exosuit. Is it a product? Can you include the relevant parameters in detail, such as stiffness, maximum output force, and so on?
Response: Unfortunately, there were no further details given. We have explained, how we have adapted the tension individually in lines 95-97. We have revised this part including your advice:
“The intensity of the elastic bands can be adjusted via loops by means of Velcro straps but cannot be scaled reproducibly regarding maximum output force. Therefore, the intensity was adjusted individually …”
Furthermore, we have discussed at the end of the Discussion:
“A limitation of the study is that the intensity of the elastic bands could not be scaled reproducibly. The integration of strain gauges into the elastic bands and pressure sensors to quantify the pressure/uncomfortableness on the body could be a further development for scientific use.”
Point 3: When subjects wearing the passive exosuit are sitting, they may be feeling uncomfortable due to squeezing pressure from the elastic bands. Can you quantify the pressure? Furthermore, is there any evaluation standard indicating that the squeezing pressure is within a reasonable range?
Response: No, we were not able to measure uncomfortableness pressure on the body in our study. We have added this aspect to our limitations and outlook:
“The integration of strain gauges into the elastic bands and pressure sensors to quantify the pressure/uncomfortableness on the body could be a further development for scientific use.”
Furthermore, we have discussed this aspect of tolerating external forces:
“… the supportive effect may even increase, since a familiarisation effect is described for tolerating external forces [31].”
Point 4: Can the opto-electronic systems be used to measure the peak angular velocity of the thigh to reflect the reliability of the inertial sensing method?
Response: Please see our reference number 20, where we have evaluated the thigh-worn sensor measurement of PAV by an opto-electronic system and force plates.
Point 5: I think the statement of Conclusion part is too short to achieve a comprehensive summary.
Response: We added further aspects to the conclusion to achieve a more comprehensive summary.
Reviewer 2 Report
sensors-2141032
Summary
This study looks at the use of an exosuit for assisting geriatric patients in the sit-to-stand transition. The authors study indicates the effectiveness of commercial exosuits for assisting patients. The sit to stand transfer is difficult for geriatric patients and their inability to perform this can lead to falls and serious injuries in many cases. Exosuits are supposed to be lighter and more wearable, lacking a rigid structural support system. The authors show that exosuits developed for heavy lifters in industrial roles can be repurposed for geriatric patients. The authors propose to use PAV, the peak angular velocity of the thighs to quantify the acceleration / stabilization of the STS.
Conclusion: The study is thorough and the applications of this technology appear to be clinically interesting. My one concern is whether this manuscript is within the scope of Sensors. The manuscript is largely not focused on the development or innovation of sensor technology. That being said, I think it would be possible to enhance the sensors content in this manuscript by adding additional details and discussion. For example, there are scientists working in the wearable electronics community who develop sensors who may be interested to know what kind of sensors are optimal for monitoring geriatric patients and preventing falls. If the authors can use their study as a basis for recommending sensor design, this result would be interesting to a broader audience that reads Sensors.
Recommendation: This manuscript is thorough and well written but requires some slightly modifications to achieve the proper scope for this journal.
I have the following detailed comments on the manuscript text and figures:
1) This sentence no lines 44-46 is a bit unclear:
“While currently, lower-limb exoskeletons for 44 gait rehabilitation are available, but still in their early stages of development [11], exoskel- 45 etons for STS support are not even that.”
2) The language of ‘stored starch-energy’ is unfamiliar to me. It appears to just mean the stored potential energy in the elastic bands. Upon a quick google search, I am not able to find similar usages of ‘starch-energy’. The authors might consider another term that is more readily understandable to a broad audience.
3) It might be helpful for the authors to provide a PAV number or range characteristic of healthy younger patients. Does the exosuit bring the PAV of the geriatric patients into this range?
4) The authors may consider providing a picture or schematic to accompany the various scenarios for the STS (bed, chair with armrests, chair without armrests, etc.). This could be helpful to many readers
5) Figure 4 does not list a time scale or show tick marks. Was this intentional? Seems like it would be better to have a plot complete with a marked axis and units on the x-axis.
6) Does the exosuit inhibit the patients’ use of other modes of assistance, such as supportive ‘walkers’, canes, or similar devices? This is a tradeoff that might be useful to discuss.
7) The conclusion section is a bit sparse. The authors should include additional discussion explaining the significance of the results to remind the readers.
Author Response
Thank you for your time and effort to read our manuscript. Your comments were very helpful to improve the manuscript.
Conclusion: The study is thorough, and the applications of this technology appear to be clinically interesting. My one concern is whether this manuscript is within the scope of Sensors. The manuscript is largely not focused on the development or innovation of sensor technology. That being said, I think it would be possible to enhance the sensors content in this manuscript by adding additional details and discussion. For example, there are scientists working in the wearable electronics community who develop sensors who may be interested to know what kind of sensors are optimal for monitoring geriatric patients and preventing falls. If the authors can use their study as a basis for recommending sensor design, this result would be interesting to a broader audience that reads Sensors.
Response: We agree and have added more aspects regarding the use of IMUs to measure the effect of an exosuit on the STS transfer.
Point 1: This sentence no lines 44-46 is a bit unclear:
“While currently, lower-limb exoskeletons for gait rehabilitation are available, but still in their early stages of development [11], exoskeletons for STS support are not even that.”
Response: We have revised this sentence:
“Currently, lower-limb exoskeletons are available only for gait rehabilitation, but they are still in their early stages of development [11]. In contrast, exoskeletons for STS support are not available at all.”
Point 2: The language of ‘stored starch-energy’ is unfamiliar to me. It appears to just mean the stored potential energy in the elastic bands. Upon a quick google search, I am not able to find similar usages of ‘starch-energy’. The authors might consider another term that is more readily understandable to a broad audience.
Response: We have revised this sentence:
“The supportive force of the exosuit is created by the elastic bands at the body´s back and buttocks that are stretched when sitting down. The stored energy in the elastic bands is released during the STS transfer.”
Point 3: It might be helpful for the authors to provide a PAV number or range characteristic of healthy younger patients. Does the exosuit bring the PAV of the geriatric patients into this range?
Response: It is not reasonable that geriatric patients perform as fast as young healthy persons, even when supported by an exosuit. The reasonability of the PAV values is discussed in the first paragraph of our discussion. Here, we have added now reference values.
“PAV values in this study including only geriatric patients are reasonable but lower than in our previous proof-of-concept study including young healthy persons, older adults and geriatric patients with a mean PAV of 124.6 °/s [20]. Since we measured thigh PAV in a supervised condition (laboratory), this can explain that PAV values were slightly faster than in the study that measured thigh PAV during daily life in healthy volunteers with a mean PAV of 70.7 °/s [21].”
Point 4: The authors may consider providing a picture or schematic to accompany the various scenarios for the STS (bed, chair with armrests, chair without armrests, etc.). This could be helpful to many readers
Response: Figure 1 already shows the setup of the different conditions. If any additional clarification is needed, please specify in more detail what we should add.
Point 5: Figure 4 does not list a time scale or show tick marks. Was this intentional? Seems like it would be better to have a plot complete with a marked axis and units on the x-axis.
Response: Yes, this was intention. We have explained in the statistics why the x-axis shows no tick marks. The time was standardised relative to the signal length. We added this point also to the figure legend.
“Filtered sensor signals were visualized with the same length by interpolation. The Dynamic Time Warping Barycenter Average method [26] was applied to average the time series. Dynamic Time Warping compensates differences between sensor signals by allowing an elastic shift of the time axis [27].”
Point 6: Does the exosuit inhibit the patients’ use of other modes of assistance, such as supportive ‘walkers’, canes, or similar devices? This is a tradeoff that might be useful to discuss.
Response: In this study we have focussed only on the STS transfer, but not on walking, because the elastic bands are no stretched during walking with this kind of exosuit. Therefore, a possible use of walking aids is not relevant. We have added in the Discussion:
“In this study we have focussed only on the STS transfer, but not on walking, because the elastic bands are no stretched during walking with this kind of exosuit. Future studies should investigate possible effects of soft exoskeletons/exosuits on walking performance.”
Point 7: The conclusion section is a bit sparse. The authors should include additional discussion explaining the significance of the results to remind the readers.
Response: We added further aspects to the conclusion to achieve a more comprehensive summary.